# Report of Two Cases of Pediatric IgG4-Related Lymphadenopathy (IgG4-LAD): IgG4-Related Disease (IgG4-RD) or a Distinct Clinical Pathological Entity?

**DOI:** 10.3390/children9101472

**Published:** 2022-09-26

**Authors:** Mariaclaudia Meli, Marta Arrabito, Lucia Salvatorelli, Rachele Soma, Santiago Presti, Maria Licciardello, Vito Miraglia, Maria Grazia Scuderi, Giuseppe Belfiore, Gaetano Magro, Giovanna Russo, Andrea Di Cataldo

**Affiliations:** 1Hematology-Oncology Unit, Department of Clinical and Experimental Medicine, G. F. Ingrassia University Hospital of Catania, AOU Policlinico “G. Rodolico—San Marco”, 95123 Catania, Italy; 2Anatomic Pathology Unit, University Department of Medical and Surgical Sciences and Advanced Technologies, G. F. Ingrassia University of Catania, AOU Policlinico “G. Rodolico—San Marco”, 95123 Catania, Italy; 3Pediatric Surgery Unit, Department of Medical and Surgical Sciences and Advanced Technologies, G. F. Ingrassia, University of Catania, AOU Policlinico “G. Rodolico—San Marco”, 95123 Catania, Italy; 4Unit of Paediatric Radiology, AOU Policlinico “G. Rodolico—San Marco”, 95123 Catania, Italy

**Keywords:** lymphadenopathy, nodes, IgG4, lymph node enlargement

## Abstract

IgG4-related disease (IgG4-RD) is a recently discovered immune-mediated fibroinflammatory condition, uncommon in the pediatric population, that could involve multiple organs and induce cancer-like lesions and organ damage. Its main features are multiple injuries in different sites, a dense lymphoplasmacytic infiltrate rich in IgG4 plasma cells, storiform fibrosis, and often high serological concentrations of IgG4. Autoimmune pancreatitis is the most common manifestation, mainly in adults. Two cases of IgG4-RD in children with lymph node localization of disease are reported. Localized or systemic lymph node involvement is common, but lymph node enlargement as the first and only manifestation of IgG4-RD is unusual, and therefore, hard to differentiate from other diseases. IgG4-related lymphadenopathy (IgG4-LAD) is most likely a distinct disease, described as isolated lymphadenopathy, related to the presence of elevated numbers of IgG4-positive plasma cells. Both disorders are likely to be misdiagnosed in children because they are characterized by rare and polymorphic features. IgG4-RD and IgG4-LAD should be considered in the differential diagnosis of disorders characterized by lymphadenopathy of uncertain etiology.

## 1. Introduction

IgG4-related disease (IgG4-RD) is a newly recognized disorder of unknown etiology and pathogenesis. It is an immune-mediated fibroinflammatory condition that may involve multiple organs and induce cancer-like lesions and organ damage [1]. Its main features include: several lesions in different areas, dense lymphoplasmacytic infiltrate with many IgG4 plasma cells, storiform fibrosis, and often high serological concentrations of IgG4. It can be confused with cancer, immune-mediated disease, or infection and it is frequently presented as Type 1 autoimmune pancreatitis. This disorder was first reported in 2001, when a correlation between autoimmune pancreatitis and increased serum IgG4 levels was discovered [2]. Moreover, many districts may be involved: the bile ducts, gallbladder, salivary glands, orbital tissues, lungs, liver, lacrimal glands, kidneys, retroperitoneum, aorta, thyroid, lymph nodes, and, rarely, the gastrointestinal tract. The disease was often incidentally found after the excision of possible malignant lesions. Due to the novelty of this diagnosis, the epidemiology of IgG4-RD remains insufficiently reported and the disease is often misdiagnosed as cancer, infection, or an immune-mediated condition. Many individuals suffering from IgG4-RD have been reported in Japan [3], whereas data regarding its prevalence in Europe or elsewhere, where it seems to be less frequent, are not available. In contrast with autoimmune diseases, which are more common in women, this disease is predominantly found in men. The pathophysiology of IgG4-RD is still unknown. Usually high levels of T helper cells and regulatory T cells are detected, most likely caused by an antigen stimulating the immune system. Therefore, interleukin (IL)-4, 5, 10, and 13 and transforming growth factor beta are produced and B-cells become activated, resulting in IgG4 expression and fibrosis [4]. A conclusive diagnosis is based on the integration of clinical features, laboratory findings, and histopathological results. Indeed, three main criteria need to be met to make an IgG4-RD diagnosis: (1) diffuse or localized edema or tumor within one or more organs; (2) high serum IgG4 levels (>1.4 g/L); and (3) histopathological pattern with abundant lymphoplasmacytic infiltration, fibrosis, presenting more than 10 IgG4+ cells/HPF (high power field), and a IgG4+/IgG+ ratio >40%. (5) A diagnosis is likely if Criteria 1 and 3 are present, or if Criteria 1 and 2 are simultaneously observed [5]. However, several patients with clinicopathologic features of IgG4-RD do not show elevated serum IgG4 levels [6].

The histopathology includes lymphoplasmacytic infiltrate, storiform fibrosis, obliterative phlebitis, and drastic IgG4+ plasma cell infiltrates; recently, a tru-cut biopsy has been informative enough to reach a diagnosis (6).

Karim et al. described IgG4-RD features in children in a systematic review that showed 25 cases of pediatric disease involvement. Most of the pediatric patients with IgG4-RD developed orbital disease (44%) and pancreatitis (12%), with other manifestations, such as cholangitis, lymphadenopathy and lung disease, being less common [7].

IgG4-related lymphadenopathy (IgG4-LAD) is a distinct disease, described as isolated lymphadenopathy enriched with IgG4-positive plasma cells. Most cases of IgG4-LADs are related to immune disorders (e.g., autoimmune lymphoproliferative syndrome, Castleman disease, and vasculitis). Non-specific lymphadenopathy with IgG4-positive plasma cells (i.e., vasculitis; Castleman disease) should be considered in differential diagnosis [8].

## 2. Case Reports

Two cases of children with lymphadenopathy associated with high levels of IgG4 plasma cells in the histological exam are reported. Both patients were admitted to the Pediatric Hemato-Oncologic Unit of the University of Catania.

The first case is about a 14-year-old Caucasian male with related parents. He complained about a 5-month left submandibular lymphadenopathy. Lymph node ultrasound results showed oval morphology, hypoechoic structure, thickened capsule, and evident central stria with a 3 cm diameter. Laboratory results showed a normal blood cell count, C-reactive protein (CRP) test, and erythrosedimentation rate (ESR); the anti-cytomegalovirus, Epstein-Barr virus, Rubeola virus, Herpes simplex virus 1 and 2, toxoplasmosis, and Bartonella IgM were absent. Head and neck computed tomography (CT) scans detected multiple enlarged lymph nodes with hypervascularity and normal central hyperechoic stria. Two courses with amoxicillin-clavulanic acid and ampicillin sulbactam, respectively, were given, with no benefit. Furthermore, a tuberculosis and thyroid function test were administered and the results were negative. Immunoglobulins had normal values compared with age, but a high level of IgG4 was present: 1.85 g/L (normal: 0.08–1.4). Therefore, the patient underwent a total body CT scan, which showed bilateral lymphadenopathy in the neck with a maximum diameter of 3 cm, whereas no pathological evidence was detected in the chest or abdomen. Excisional biopsy of the neck lymph nodes revealed preserved lymph node architecture with reactive follicular hyperplasia, and immunohistochemical studies showed normal distribution of B cells (CD20-positive) and T cells (CD3-positive). No immunoreactivity for BCL-2 in germinal centers of lymphatic follicles was observed. Instead, there was marked intrafollicular but also interfollicular plasmacytosis, with an elevated number of polyclonal plasma cells. This was detected and confirmed by immunohistochemistry for kappa and lambda light chains, especially for IgG4 heavy chains (absolute number of IgG4 + cells 40–50/HPF; percentage of cells IgG4+/IgG+ about 40% in germinal centers) (Figure 1). Castleman disease was ruled out due to negative search for HHV. Based on these clinical, serological, morphological, and immunohistochemical features, a diagnosis of type II (reactive follicular hyperplasia-like) IgG4-related lymphadenopathy was reached. No treatment was administered. Clinical and ultrasound follow-up was performed for 8 years, with normalization of serum IgG4 level and no relapse and/or new organ involvement.

The second case concerns a 16-year-old boy with negative family history and a psychomotor delay. He presented in the emergency room because of two episodes of loss of consciousness and muscle tone with spontaneous resolution. His brain CT was negative. Laboratory results showed high acute-phase reactants (ESR 110 mm/h, CRP 176.4 mg/dL, normal range 0–5 mg/dL) and anemia (Hb 8.9 g/dL). The abdominal ultrasound highlighted hepato-splenomegaly and detected some hypervascularized lymph nodes that were increased in volume (maximum diameter 5 cm) in the mesentery and mesogastric regions. Upon admission to our unit, the patient was in good general condition and reported no weight loss, sweating, or other noteworthy symptoms. Normocytic anemia was confirmed whereas hepatic and renal function blood tests, viral tests, celiac disease screening, fecal occult blood, and fecal calprotectin tests were negative, and ALPS was excluded. Electroencephalogram (EEG) showed a pathological pattern, with generalized tip/wave discharges; although the brain MRI was unremarkable, anti-epileptic therapy with Levetiracetam was administered. A neck-chest-abdomen CT scan confirmed multiple lymphadenopathies in the abdomen. A tru-cut biopsy revealed nonspecific hyperplastic lymphadenitis; subsequently, the patient underwent excisional biopsy of the biggest lymph node (Figure 2A,F). Histology results showed remarkable and diffuse follicular hyperplasia, sometimes with atretic germinal centers traversed by penetrating vessels—lollipop follicles—and thickened mantle zones with lymphocytes arranged in layers—onion skin appearance—simulating Castleman disease. Rare progressively transformed germinal centers, focal intranodal fibrous septa, and aspects of capsular venulitis were observed. Interfollicular zones were expanded by numerous polyclonal plasma cells (Figure 2H,I). Immunohistochemical studies revealed normal distribution of B cells (CD20-positive) and T cells (CD3-positive). No immunoreactivity for BCL-2 in the germinal centers of lymphatic follicles was detected. A significant number of interfollicular plasma cells (CD138-positive; (Figure 2D,G) (>40/HPF) was also positive for IgG4 (Figure 2L). Two diagnostic hypotheses were considered, namely a multicentric plasma cell variant of Castleman disease (clinical presentation: hepatosplenomegaly; morphological features of Castleman-like disease) or IgG4-related lymphadenopathy. IgG4 levels turned out to be elevated (1.98 g/L, normal: 0.08–1.4), whereas immunohistochemical analyses for HHV8 (specific for Castleman disease) was negative, thus leading to the final diagnosis of IgG4-related lymphadenopathy. The patient was discharged and put on a “wait and see” follow-up with ultrasound surveillance; no therapy was prescribed. The patient is currently healthy after four years follow-up.

## 3. Discussion

IgG4-RD is rarely detected in the pediatric population, with average age at diagnosis being over 50 years [9,10,11]. Pancreatic involvement is less common in children than in adults and most pediatric patients complain of other symptoms, such as orbital involvement followed by sinunasal impairment, lacrimal gland disorder, and sialadenitis [12]. Other localizations include the mediastinum, retroperitoneum, lung, and mesenterium, as reported in the systemic literature studies conducted by Karim et al. [7]. Localized or systemic lymph node involvement is common [7,13,14], although cervical lymph node expansion as the initial and only manifestation of IgG4-RD is unusual and often hard to differentiate from other diseases (Case 1). Wu et al. presented a case of IgG4-RD manifesting as cervical lymphadenopathy that was treated with oral steroid therapy and showed a complete regression in one week with no recurrence after three months follow-up [15]. Usually, multiple lymph nodes are involved in the mediastinum, abdomen, and axillae. Corujeira et al. presented a 22-month-old girl with recurrent respiratory tract infections whose thoracic CT and MRI detected several mediastinal lymphadenopathies. Histology results revealed lymphoplasmacyte infiltrate, increased IgG4-positive plasma cells, and above 40% ratio of IgG4/IgG-positive cells. The girl was treated with glucocorticoids and achieved symptomatic improvement, volume mass decrease, and reduction of serum IgG4 levels after 6 weeks [16]. However, elevated serum IgG4 levels are neither necessary nor sufficient to diagnose IgG4-RD and, in cases of low serum IgG4 levels with high clinical suspicion for the disease, it is required to verify the presence of a prozone effect [17]. Asymptomatic IgG4-LAD has also been incidentally identified through radiologic images or unexpectedly detected in pathological specimens (Case 2). Cheuk et al. reported a case of an asymptomatic patient with a lymphadenopathy incidentally found during an investigation for aortic stenosis [14].

Differential diagnosis includes several disorders due to the wide histological spectrum of IgG4-related lymphadenopathy [18,19,20].

The main differential diagnosis is with Castleman disease [20], which includes several distinct lymphoproliferative disorders with different pathogenesis and clinical manifestations. Castleman disease can be considered a unicentric or multicentric disorder. The former is characterized by single lymph node involvement and usually occurs in the mediastinum, cervical region, or abdominal/pelvic cavity. The second form, in which a Herpesvirus-8 infection is often observed [21,22,23], exhibits involvement of multiple lymph node stations and displays inflammatory symptoms, cytopenias. Such as autoimmune hemolytic anemia, liver or kidney dysfunction, and peripheral neuropathy [23]. Although these last symptoms aren’t typical of the IgG4-RD disease, the presence of tissue eosinophilia and involvement of cervical lymph nodes, eyes, salivary glands, or the pancreas could support the diagnosis of multicentric Castleman disease. However, the expansion of the interfollicular area, follicular hyperplasia, and gradual transformed germinal centers in which the number of IgG4-positive plasma cells and serum IgG4 levels are not increased, allow for the exclusion of Castleman disease [24,25].

Rosai-Dorfman disease should also be considered in the differential diagnosis between IgG4-RD with lymph node involvement and IgG4-LAD. It is an uncommon histiocytic disorder. As IgG4-RD, cervical lymphadenopathy is the main presentation, and extranodal involvement in the head and neck, intracranium, bone, heart, skin, parotid gland, periodontium, orbit, thyroid, breast, and paranasal sinuses has also been detected [26,27]. It is self-limiting, rarely life-threatening, and treatment is not often required, similar to our IgG4 cases. These two diseases are also similar in histological patterns. Indeed, the inflammatory infiltrate is mixed, including plasma cells, histiocytes, and occasionally eosinophils, with a variable fibrosis, mimicking IgG4-related lymphadenopathy [28,29,30]. Furthermore, an elevated number of IgG4-positive plasma cells is also observed in Rosai-Dorfman disease [30], but a noteworthy marker is the immunoreactivity for S100 protein in histiocytes expressed in Rosai-Dorfman disease, which can be used to differentiate the two disorders [31,32,33].

Sjogren’s syndrome should also be considered in the differential diagnosis. It is a generalized autoimmune disorder, and its main feature is a lymphocytic infiltrate of the exocrine glands, causing dysfunction and organ damage. The salivary and lacrimal glands are mostly involved, leading to dry eyes and mouth, but general organ impairment is possible, manifesting in anemia, arthralgia, sensory neuropathies, respiratory disfunction, and interstitial nephritis [33]. In the pediatric population, recurrent parotitis is a usual manifestation of this disease. Diagnosis relies on laboratory results: anti-Ro anti-La antibodies within normal limits exclude the disease.

Finally, IgG4 disease should be differentiated from sarcoidosis, a multisystemic granulomatous disease. This disorder is characterized by intrathoracic involvement with symmetrical bilateral hilar adenopathy and diffuse lung micronodules; however, extrapulmonary manifestations such as skin lesions, uveitis, liver or splenic failure, and peripheral arthritis have also been reported [34]. Furthermore, peripheral lymphadenopathy involving the cervical, axillary, and inguinal node stations may be present, but the discovery of noncaseating granulomas should be specific enough to differentiate the disorder.

All of these diseases present multi-organ and aspecific symptoms, often similar to one another; therefore, a distinct diagnosis requires a histological examination.

Different types of treatment for IgG4-RD have been proposed. Patients with subclinical lymphadenopathy or mild submandibular gland enlargement can be managed with a “wait and see” approach [6], as in the cases reported herein. Spontaneous remission, or temporary remission of IgG4-RD without treatment have been described [33]. Steroids are the first choice treatment for symptomatic patients. Prednisone at a dosage of 30–40 mg/day is the usual initial form of therapy and is administered for 2–4 weeks and tapered off in 3–6 months [6]. A review conducted by Karim et al. reports that prednisone as the only treatment was successful in only 43% of cases [7]. The doses of administered prednisone were not detailed for every case included in the review, but when reported, it was usually between 0.5 and 2 mg/kg/day. Nineteen of the twenty-three cases treated with prednisone showed early remission [7]. Methotrexate, micophenolate mofetil, and azathioprine were used in 50% of patients as steroid-sparing drugs or remission-maintenance agents following steroid-induced recovery. Rituximab appears to be a strong alternative in case of no response [10,11,12,13,14]. Nonetheless, a long follow-up is advisable to identify possible relapses early. The early diagnosis and therapy, particularly for the advanced subtypes of the disease, are crucial for preventing severe and irreversible effects. If left untreated, this disease can lead to fibrosis and organ damage, such as chronic kidney disease and pancreatic and liver failure [35]. However, no complications or relapses were reported in our cases, although the young ages of the two patients at disease onset should be noted.

The presentation of IgG4-LAD could also be related to extra-nodal lesions resulting in IgG4-RD or as an isolated lesion with a clinically and biologically distinct benign and non-progressive behavior. Up to now, the association between IgG4-RD and IgG4-LAD is still not clearly described [1,8,36]. 

A 7-year retrospective analysis (2012–2018), conducted by the Italian Paediatric Haematology and Oncology Association (AIEOP), searched over 832 nodal examinations and described nine cases of LAD [8]. Every patient was of Caucasian origin, with a median age of 15.4 years at diagnosis and with greater involvement of men than women [8]. Cases of IgG4-LADs associated with immune regulation or autoimmune disorders were described [7,8].

The “wait and see” strategy is the most commonly used for stable cases of IgG4-LAD; however, some patients received steroid treatment resulting in early complete clinical remission [8].

In conclusion, part of IgG4-LAD may constitute a precocious phase of IgG-RD, or IgG4-LAD may be a different entity altogether. We suggest that new IgG4-LAD patients should undergo close clinical monitoring.

Furthermore, we would like to underline the relevance of including IgG4-LAD and IgG4-RD in the differential diagnosis of lymphadenopathy of uncertain etiology. Although this disease is uncommon in the pediatric population, knowledge and awareness of it could be pivotal in avoiding misdiagnosis and delaying possible care, particularly in childhood.

## Figures and Tables

**Figure 1 children-09-01472-f001:**
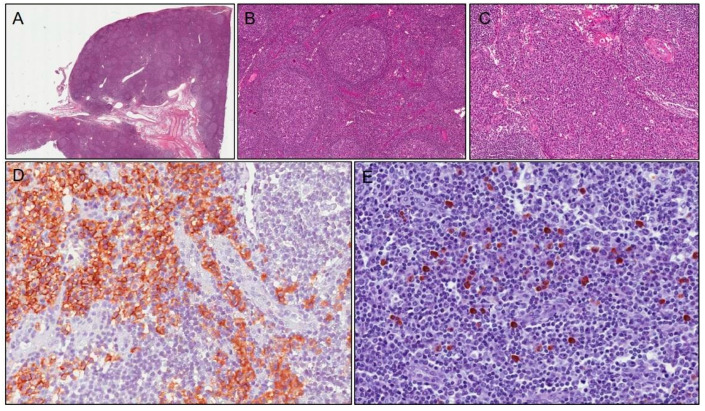
Case 1: histological examination shows follicular hyperplasia (**A**); with thickened mantle zones and lymphocytes arranged in layers—an onion skin appearance (**B**); there are numerous plasma cells in interfollicular area (**C**) stained with CD138 (**D**). Immunoreactivity for IgG4 shows positivity in about 40 plasma cells/HPF (**E**).

**Figure 2 children-09-01472-f002:**
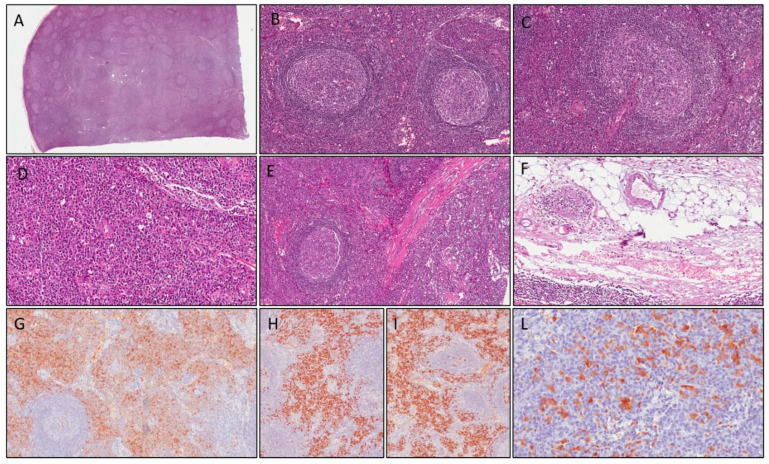
Case 2: histological examination shows follicular hyperplasia (**A**); thickened mantle zones with lymphocytes arranged in layers—an onion skin appearance (**B**); atretic germinal centers traversed by penetrating vessels—lollipop follicles (**C**); numerous polyclonal plasma cells in interfollicular area (**D**); intranodal fibrous septa (**E**) and aspects of capsular venulitis (**F**). Plasma cells in interfollicular area are stained with CD138 (**G**) and show polyclonal profile (positive for kappa and lambda light chains, respectively, in (**H**,**I**). Immunoreactivity for IgG4 was positive in >40 plasma cells/HPF (**L**).

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
