# Peer review of "Report of Two Cases of Pediatric IgG4-Related Lymphadenopathy (IgG4-LAD): IgG4-Related Disease (IgG4-RD) or a Distinct Clinical Pathological Entity?"

_children, 2022, doi:10.3390/children9101472_

Round 1
Reviewer 1 Report (Previous Reviewer 1)
I do think your case report highlights interesting patients and the difficulty of diagnosting IgG4-related disease. However, the translation to English is difficult to decipher at times and makes the overall report difficult to read. I'm not sure what editing services available to you, but I do believe this needs to be addressed prior to publication.
Author Response
Dear Reviewer,
Thank you for your suggestion.
We made extensive revisions in terms of English as you suggested
Reviewer 2 Report (New Reviewer)
Dear authors,
thank you for your valuable work and effort. It is an interesting work that will help clinicians in the future with this under-recognized entity. Here are my comments:
The manuscript needs extensive revision in terms of English language. Moreover, the title of the article does not reflect the content. Please spare the reference to IgG4-RD for the introduction and discussion sections. The differential diagnosis in the end is useful, but is beyond the scope of this paper (e.g. lines 219-221 for the treatment of multicentric Castleman disease).
Line 54: the term "organ system" does not stand for all sites listed (e.g. retroperitoneum)
Line 79: pls rephrase "increasingly small biopsy samples that frequently"
Line 86: abbreviation for ALPS
Line 118 vs 130: IgG4+ is 40 or >50%?
Lines 124-125: ultrasound follow-up? subsequent serum IgG4 levels?
Line 135: it is important to provide the cut-off positivity limit for CRP
Line 142: immune etiology of seizures are associated with the presence of continuous slow waves, frontal intermittent rhythmic delta activity, and delta brush in EEG -pls specify EEG pathological findings
Lines 278-279: According to Ref #34 the abundance of IgG4+ plasma cells in isolated reactive lymphadenopathy is no indication of IgG4-RD. Please comment on the findings of this study
According to Ref #8 "pediatric IgG4-LAD may (thus) represent one of the possible clinical manifestations of ALPS". Pls comment this observation
A relevant review for IgG4-RD (i.e. https://doi.org/10.1080/1744666X.2017.1354698) should be included in text, as it mentions that "elevated IgG4 serum level is neither necessary nor sufficient to diagnose IgG4-RD and in cases of low IgG4 serum level with high clinical suspicion for the disorder, one must check for the presence of prozone effect", "measuring circulating plasmablasts or IgG4/IgG RNA ratio may be superior biomarkers in active IgG4-RD than measuring serum IgG4 level" and "18F-FDG PET/CT imaging patterns strongly suggestive of IgG4-RD".
In view of a recent publication (DOI: 10.3390/diagnostics11122213) isolated IgG4-LAD has unique ultrasonographic features: cortex was filled with diffusely scattered hyperechoic foci and some bright foci gathered to form a follicle ("starry night sign"). Does this finding agree with yours?
According to another recently published paper (https://doi.org/10.1007/s00296-021-04885-5) namely a retrospective study on 6 children with IgG4-RD and 2 children with IgG4-LAD reported orbital involvement in all 6 patients with IgG4-RD (with swelling). Another interesting review of pediatric cases is also missing from the discussion (https://doi.org/10.1007/s10067-017-3934-9).
Thank you.
Author Response
Dear Reviewer,
Thank you for your comments.
We made a revision of the English language as you suggested.
We have also made the following corrections you suggested: line 54, line 79, line 86, line 135, 118 and 130.
line 142: EEG features are: generalized tip/wave discharges. We reported the description in the text.
There was a normalization of serum level of IgG4 after the excisional biopsy and a ultrasound follow-up was also performed. I added these points in the text.
I added the references you suggested.
No starry night sign was described in the ultrasounds performed.
We used this title because IgG4-RD with lymph node involvement have been described in the literature and these can be confused with IgG4 LAD which appears to be a distinct pathology. However it is not clear, to date, whether there is a correlation between this two diseases or if they are two distinct entities.
Round 2
Reviewer 1 Report (Previous Reviewer 1)
I like your paper and I think your described patients and subsequent discussion are relevant for this difficult diagnosis. I realize that this paper has been translated from your native Italian to English and appreciate that you have made efforts to revise the manuscript for better readability and flow.
Author Response
Dear Reviewer
we made new English language changes as you suggested
Reviewer 2 Report (New Reviewer)
Dear Authors,
thank you for your response and the improvements you made. Still the manuscript needs refinement in terms of English language usage. I expected the manuscript to be polished of minor errors during the reviewing process, but you need to proof-read it thoroughly (e.g. 1,98 g/L in line 154). My personal opinion is that both entities in the title might link, but no clear pathogenetic mechanism has been displayed, so the term "onset" does not stands well for me. You have answered to all of my enquires, so thank you again.
Author Response
Dear Reviewer,
thank you for your comment.
We made English language changes as you suggested and we modified the word "onset" in the title with "part".
This manuscript is a resubmission of an earlier submission. The following is a list of the peer review reports and author responses from that submission.
Round 1
Reviewer 1 Report
I think this paper shows a good/reasonable knowledge of the subject matter and highlights a sound need for evaluation of this elusive disease in a younger population.
If I could offer any minor suggestion, it may be to expound a bit more on the entities within their discussed differential diagnosis, particularly what the 'Castleman-variant of IgG4-related lymphadenopathy' is and how that may differ clinically regarding treatment and further disease progression. Other entities mentioned, similarly, were given less attention - for instance, how would Rosai-Dorfman or Sjogren's syndrome or sarcoidosis differ clinically from IgG4-related LAD (are they identical?) I felt these entities were merely mentioned as part of the differential diagnosis. Differentiating reasons may be apparent to the authors, but should be explained a bit more I think.
Another discussion point would be regarding treatment - how do these other entities discussed differ regarding treatment and, if left untreated, what 'grave sequelae' (a term used in the 2nd to last paragraph) may befall the pediatric patient diagnosed with IgG4-RD? Autoimmune pancreatitis is described for adult disease in the intro - does this occur in children as well? The cases discussed showed no clinical damage to the patient, who remain in good health despite their diagnoses. Are there truly 'grave sequelae' for the pediatric patient? There may be no answer for that - which may be left as an 'unknown'... or perhaps there is risk for later adult disease(s)? Alternatively, a positive for children diagnosed with IgG-RD is avoiding a misdiagnosis which may involve subsequent unnecessary and damaging treatment (this IS mentioned as an alternative to treatment in the clinically unaffected patient = 'watchful waiting').
Overall, I thought the paper offered a straightforward design describing individual cases and offered a brief lit review on a subject that is not very well understood (even in adults). I found no elements of bias and I do feel the paper is well-written and easy to follow.
Author Response
Dear reviewe
Thank you very much for your comments.
How you suggest, I expanded the description of the diseases which have similar clinical manifestations (Castelman disease, Rosai Dorfman, Sjogren Syndrome and Sarcoidosis). However I underline that the differential diagnosis with this diseases is possible just with the histological examination.
No complications or recurrent disease were reported in our patients after a long follow up (4 and 8 age). However the onset of the disease was in young patients and more time is needed for a answer to your question "are truly grave sequele for the pediatric patient?". However no significant data where reported in literature for pediatric patients' outcome
Reviewer 2 Report
This is a case report of two children with lymphadenopathy that show increased numbers of IgG4+ plasma cells / HPF .
There are a number of issues with this report
1. The most recent consensus diagnostic features (Wallace Z et al, 2020) were not used. This is important as lymph node is not one of the 11 key sites involved (salivary glands, lacrimal gland, orbit, lung, kidney, aorta, retroperitoneum, pancreas, biliary, thyroid and meninges). Although lymph nodes can be involved, these are not a key site and lymph nodes containing numbers of IgG4+ plasma cells can be seen in many conditions, including other autoimmune disease such as Sjogren’s, SLE, Castleman’s and even sporadic reactive lymph nodes (Cheuk and Chan, 2012). Cheuk and Chan make the point that increased numbers of IgG4+ plasma cells especially the low numbers (40-50/HPF) of the cases in this paper can be seen in reactive lymphadenopathy. Usually where lymph nodes are examined in patients who have confirmed IgG4-RD in any of the 11 key sites have >100 IgG4+ plasma cells/ HPF.
2. The authors are not up to date with the epidemiology pathogenesis of IgG4RD and I would disagree that the epidemiology is poorly described. There are many more papers regarding pathogenesis since the article cited from 2015. The authors do not cite the review by Karim et al of IgG4-RD in pediatrics.
3. English needs editing. Line 61 – parents had consanguinity.
4. There is no follow-up clinical data given for patient 1. Was there any resolution of the organomegaly? Patient 2 had reduction of the lymphadenopathy without treatment which is very unusual for IgG4RD.
5. Was HHV8 immunohistochemistry done to exclude Castleman’s disease, HHV8-related subtype?
6. The references given are not for children. Wu et al described lymph nodes in a 63 year old man with bilateral submandibular and parotid involvement – the salivary glands are key sites of involvement. Cheuk et al described cases in adults. Corujera described lung, mediastinal and lymph node involvement.
7. The authors do not provide a literature review as indicated in the title and do not discuss the consensus view on occurrence of IgG4RD in children eg reffs
8. The elevation of the serum IgG4 in both cases is quite minimal. Not even twice the upper limit of normal.
Author Response
Thank you for your comments, I answer in the attached file.
Regards

Round 2
Reviewer 2 Report
The authors have made some changes as suggested by the reviewers. However, they are still using outdated criteria for the diagnosis of IgG4-RD (Wallace et al, 2020). The authors’ cases only have lymphadenopathy and this is not one of the 11 sites required for the diagnosis. Referring to cases that were published as IgG4-RD which would not fulfil the criteria now accepted for IgG4-RD is not helpful.
The authors should read again the article by Cheuk and Chan, 2012 about the under- and over-diagnosis of IgG4-RD lymphadenopathy. They state that “ in the absence of well documented IgG4-RD, a diagnosis of IgG4-related lymphadenopathy should be made with great caution.” “A descriptive diagnosis of reactive lymphoid hyperplasia with increased iGg4+ plasma cells should be made” .
The numbers of IgG4+ plasma cells in the lymph nodes in these 2 cases are less than recommended for lymph node involvement – seen where the patient has known IGG4_RD.
The two cases presented had only lymph nodes involved and no specific IgG4-RD site involvement was noted at 4 years and 8 years followup. There is no supportive evidence for the diagnosis of IgG4-RD.
Author Response
Dear Reviewer
Based on your suggestions we modified the title and the text according to IgG4 lymphadenopathy disease (IgG4-LAD) features' used by Pillon et al (Pediatric IgG4-related lymphadenopathy: a rare condition associated with autoimmunity and lymphoproliferative disorders)
Our histological exames were revisioned by the national referent anatomopathologist
Round 3
Reviewer 2 Report
1. The total number of IgG4+ plasma cells / HPF is 40-50 in these two cases. – not >100/ HPF as reported in the pediatric study by Mainard et al or the two adult studies by Cheuk and Chan. This is insufficient for a histological diagnosis of lymphadenopathy with increased numbers of IgG4+ cells.
2. The authors persist in maintaining the diagnosis is based on 3 features from the consensus 2012 paper without inclusion of discussion of the Wallace et al consensus criteria.
3. The abstract again states that this is IgG4-RD rather than lymphadenopathy with increased numbers of IgG4+ cells.
4. The pediatric paper by Mainard et al highlighted that none of their 9 patients lymphadenopathy with increased numbers of IgG4+ cells developed IgG-RD after followup and in fact one developed nodular lymphocyte predominant Hodgkin lymphoma, two developed ALPS and one had other autoimmune disorders. Similarly in the adult series, increased numbers of IgG4+ plasma cells have been reported in a variety of disorders – rheumatoid arthritis, Sjogren’s, Rosai Dorfman, Castleman’s and even sporadic reactive lymph nodes. Line 69 – the authors state most cases of lymphadenopathy with increased numbers of IgG4+ cells are associated with Immune disorders – but no reference is given.
5. There are cohort studies outside Japan – line 46 – see 2020 paper by FLoreani but I agree that there needs to be more studies but consensus criteria must be used for the diagnosis.
6. Neither patient developed other organ involvement after 4- 8 years of followup and neither progressed without treatment which is unlike IgG4-RD.
7. A lot of the discussion is redundant – the differential diagnosis of sarcoid is clinical only as prominent granulomas is an exclusion criterion for IgG4-RD